# Uncovering the causes and socio-demographic constructs of stillbirths and neonatal deaths in an urban slum of Karachi

Ameer Muhammad[1]*, Muhammad Salman Haider Rizvee[2], Uzma Khan[1], Hina Khan[1], Alishan Bachlany[1], Benazir Baloch[3], Yasir Shafiq[4,5,6,7,8]

**1** VITAL Pakistan Trust, Karachi, Pakistan, **2** Medical College, The Aga khan University, Karachi, Pakistan, **3** Department of Pediatrics and Child Health, The Aga Khan University, Karachi, Pakistan, **4** Centre of Excellence for Trauma and Emergencies (CETE) & Community Health Science, The Aga Khan University, Karachi, Pakistan, **5** CRIMEDIM–Center for Research and Training in Disaster Medicine, Humanitarian Aid, and Global Health, Università del Piemonte Orientale, Novara, Italy, **6** Department of Translational Medicine, Università del Piemonte Orientale, Novara, Italy, **7** Department of Pediatric Newborn Medicine, Brigham and Women's Hospital, Boston, MA, United States of America, **8** Department of Global Health and Population, Harvard T.H. Chan School of Public Health, Boston, MA, United States of America

☯ These authors contributed equally to this work.

* ameer.muhammad@vitalpakistantrust.org

**Data Availability Statement:** All relevant data are within the manuscript and its Supporting Information files.

## Abstract

### Introduction

Neonatal deaths and stillbirths are significant public health concerns in Pakistan, with an estimated stillbirth rate of 43 per 1,000 births and a neonatal mortality rate of 46 deaths per 1,000 live births. Limited access to obstetric care, poor health seeking behaviors and lack of quality healthcare are the leading root causes for stillbirths and neonatal deaths. Rehri Goth, a coastal slum in Karachi, faces even greater challenges due to extreme poverty, and inadequate infrastructure. This study aims to investigate the causes and pathways leading to stillbirths and neonatal deaths in Rehri Goth to develop effective maternal and child health interventions.

### Methods

A mixed-method cohort study was nested with the implementation of large maternal, neonatal and child health program, captured all stillbirths and neonatal death during the period of May 2014 till June 2018. The Verbal and Social Autopsy (VASA) tool (WHO 2016) was used to collect primary data from all death events to determine the causes as well as the pathways. Interviews were conducted both retrospectively and prospectively with mothers and caregivers. Two trained physicians reviewed the VASA form and the medical records (if available) and coded the cause of death blinded to each other. Descriptive analysis was used to categorize stillbirth and neonatal mortality data into high- and low-mortality clusters, followed by chi-square tests to explore associations between categories, and concluded with a qualitative analysis.

**Funding:** This project is funded by Bill & Melinda Gates Foundation. The grant reference number is OPP1160891. The funding was reveievd on November 2016. The funders had no role in study design, data collection and analysis, decision to publish, or preparation of the manuscript.

**Competing interests:** The authors have declared that no competing interests exist.

## Results

Out of 421 events captured, complete VASA interviews were conducted for 317 cases. The leading causes of antepartum stillbirths were pregnancy-induced hypertension (22.4%) and maternal infections (13.4%), while obstructed labor was the primary cause of intrapartum stillbirths (38.3%). Neonatal deaths were primarily caused by perinatal asphyxia (36.1%) and preterm birth complications (27.8%). The qualitative analysis on a subset of 40 death events showed that health system (62.5%) and community factors (37.5%) contributing to adverse outcomes, such as delayed referrals, poor triage systems, suboptimal quality of care, and delayed care-seeking behaviors.

## Conclusion

The study provides an opportunity to understand the causes of stillbirths and neonatal deaths in one of the impoverished slums of Karachi. The data segregation by clusters as well as triangulation with qualitative analysis highlight the needs of evidence-based strategies for maternal and child health interventions in disadvantaged communities.

## Introduction

Annually, approximately 2.5 million stillbirths and nearly 2.4 million infant deaths within the first 28 days of life are reported globally, highlighting significant yet under-addressed issues in maternal and neonatal health [1]. Around 98% of these deaths are contributing by low- and middle-income countries (LMICs), especially in sub-Saharan Africa and South Asia [2]. Most of these deaths could have been prevented with better healthcare and interventions [3, 4]. However, burden remain in high in most disadvantaged in urban slums and rural areas, where poverty, inadequate sanitation, and limited access to healthcare increase the risk of these event [5, 6].

Despite these immense challenges, significant progress has been made in reducing stillbirth and neonatal deaths worldwide. Efforts such as the Sustainable Development Goals (SDGs) and Every Newborn Action Plan are devoted to improving maternal and child health [7]. To meet the SGDs targets, countries have adopted initiatives to improve avert perinatal, neonatal and child deaths through comprehensive and quality provision of antenatal care, skilled birth attendance, emergency obstetrics, and newborn and childcare [8]. But still many LMICs are still falling behind and therefore, reaching the last mile for maternal, neonatal and child health outcomes (MNCH) is still very challenging for many of these countries [9, 10]. These challenges are mainly related to both health system as well as community factors [9, 10]. Health system related factors are mainly suboptimal care, poor infrastructure, lack of access to care, and inadequate governance. On the other hand community factors include financial constraint due to poverty, lack of education and poor health seeking behavior [9, 10].

Pakistan remains among the top ten countries with highest stillbirth and neonatal death rates. According to the latest available data, the stillbirth rate in Pakistan is estimated to be approximately 43 per 1,000 births, that is, approximately 1 in 23 pregnancies in Pakistan resulted in stillbirth [11, 12]. In terms of neonatal mortality, Pakistan has made some progress in recent years but still faces considerable challenges. The latest estimates suggest that the neonatal mortality rate in Pakistan is approximately 46 deaths per 1,000 live births, that is, nearly 1 in 22 newborns in Pakistan die within the first 28 days of life [11, 13]. It is estimated that

infants are at greatest risk of mortality during their first month, with preterm birth complications, asphyxia, and neonatal sepsis being the leading cause of death (CoD) [11] [14, 15]. Stillbirths, on the other hand, are predominantly caused by antepartum hemorrhage, obstructed labor, and pregnancy-induced hypertension [11, 15]. Factors such as poor nutrition, inadequate antenatal care, limited access to skilled birth attendants, and substandard living conditions as well as poor health seeking behaviors contribute to the high numbers of stillbirths and neonatal deaths in countries like Pakistan [16, 17].

The integrated management of pregnancy and childbirth assure the community-based antenatal care, skilled delivery, and home-based newborn care and aims to improve access to comprehensive maternal, neonatal and child health (MNCH) and avert the risk of poor pregnancy and birth outcomes like stillbirths and neonatal deaths [18, 19]. However, even with scaling up such intervention package, the SDG target of 12 newborn deaths per 1000 live births by 2030 for example is hard to achieve with existing contributing risk factors.

In depth understanding of each CoD and better knowledge about root causes may provide an opportunity to implement more robust approaches and improve the quality the continuum of the MNCH care. This paper aims to present the causes as well as socio-demographic characteristics of these events, and community-and health system-related factors leading to stillbirths and neonatal deaths at target setting.

## Methods

### Study design

This is mixed-method cohort study, captured all stillbirths and neonatal deaths during the period of 14 May 2014 till 15 June 2018 through primary data collection. In-depth data collection to determine the CoD and capture all the information related to the events comprised of both retrospective as well as prospective approach using routine community surveillance system at the study setting. [20] All interviews were conducted with mothers or caregivers. Data collectors were instructed to be careful regarding sensitive questions when conducting interviews.

### Setting

**Rehri goth.** Rehri Goth is one of the oldest coastal slums located on the Malir district Karachi, Pakistan. According to 2014 census data, the population of Rehri Goth was estimated to be approximately 42,980 [21]. Annually, the birth cohort number is approximately 1,200 [21]. Aga Khan University's Department of Pediatrics and Child Health established a surveillance site within this community to collect data on essential MNCH indicators. Baseline data from 2013–14 estimated an under-five mortality rate (U5MR) of approximately 108.6 to 92.5 per 1,000 live births, with a neonatal mortality rate (NMR) of 59.7 to 42.2 per 1,000 live births [21, 22]. The is divided into 42 clusters, each containing 250–300 households.

**Integrated MNCH services at Rehri Goth.** In 2014, the VITAL Pakistan Trust implemented integrated package of MNCH interventions (the Caplow project)at Rehri Goth in recognition of the high under-five mortality [23]. A holistic and multifaceted approach enabled the project to provide a full continuum of care to women through including antenatal, skilled delivery, nutrition and assistance in access to family planning counseling and provision of services [24]; as well as to the children under-five years of age through community newborn and child care, nutritional counseling interventions, immunization, primary health care services, and referral system [23]. Additionally, transportation assistance was provided to pregnant women and enhanced links were established with existing healthcare facilities to ensure access to timely referral and coordination of MNCH services in Rehri Goth [24]. As a part of this

work, VITAL built a collaboration and signed a memorandum of understanding with Koohi Goth Women Hospital (KGWH), a secondary care hospital at distance of 6–7 kilometers from Rehri Goth. Pregnant women registered in the MNCH program, were transported to KGWH at the time of labor. During referral, the patient was carrying an antenatal file which has unique ID for tracking the deidentified patient record if needed.

For surveillance purpose, the area is geographically divided into 42 blocks each comprises of 250–300 households. Among these blocks, nearly half has the fishermen community. In these blocks, the U5MR is as high as 200–250 per thousand live births compared to non-fishermen clusters, U5MR is below the national average of Pakistan, i.e., 79 per 1000 live births. Therefore, to track the overall progress of the project, the study site was divided into high-mortality clusters (U5MR equal to or greater than 79—national average of Pakistan), and low-mortality clusters (less than 79 per 1000 thousand live births).

**The verbal and social autopsy (VASA) tool.**   All retrospective and prospective stillbirth and neonatal death cases were captured using the combined Verbal and Social Autopsy (VASA) tool [25]. This tool was developed by the Maternal and Child Epidemiology Estimation Group (MCEE), formerly known as the Child Health Epidemiology Reference Group (CHERG) and experts from the Johns Hopkins Bloomberg School of Public Health to investigate the causes and determinants of child mortality in developing countries with higher rates of child mortality [25]. Studies has been conducted where authors used VASA tool in their context and reported the causes as well as pathways to under-five deaths [26–28], but only one study is conducted on validation of the tool (unpublished) [29] The VASA tool includes the original WHO 2016 Verbal Autopsy Questionnaire (to determine the biological cause of death) [30], in addition to a social autopsy questionnaire to understand the social, behavioral, and health system determinants of mortality [31].

In order to capture the more holistic information about the death event, this study has included open-ended questions to capture the maternal/caregiver history and narrative on other key aspect such as the detail narrative of the event, decision-making process at household, experiences surrounding the fatal event, and perception of healthcare providers' roles. Further, the narrative also included capturing the experience of health care workers and the midwives if they have any information about the fatal event.

Their views and narrative are considered to be important because they were involved tin coordinated care during antenatal period, counseling the families, arranging referral during intrapartum and postnatal period. The addition of this narrative helped data triangulation and in-depth understanding of events.

## Sample size and sampling technique

A total of 421 stillbirths and neonatal deaths were reported during the study period, and all of them were approached for VASA interview. Surveillance teams once identified the death, logged it in the registry and that information is passed to VASA team for the visit. Further, out of total, VASA interview was conducted retrospectively on 260 events, all the information was already available to the VASA team. Although these events were captured in the surveillance system in real-time, VASA study was introduced at later stage of the program. In the statistical analysis, the data of all events with 'completed visit status' is included. Furthermore, a convenient sampling was used to select a total of 40 qualitative narratives for analysis out of total narrative recorded. This approach was adopted purposefully to perform rapid and in-depth assessment of qualitative findings for triangulation with VASA data. For more representation, 20 interviews were chosen from each cluster with further segregation of 10 stillbirths and 10 neonatal deaths from each cluster.

## Data collection

The VASA tool is translated into Urdu and then back into English to ensure accuracy. A total of 3 teams were deployed, each team consisting of one senior research assistant (SRA), who lead the VASA interviews and a community health worker (CHW) who assisted and liaised with the households and community members. All the team members were multilingual and able to community communicate with the community in all local languages. The study team completed a two-week training module that included classroom and field training. Classroom training topics included quantitative and qualitative data collection techniques, interview skills, informed consent, research ethics, sensitivity and communication skills, and interview role playing. The interviews were conducted by the SRA in the local language. Paper-based form was used and all the interviews to capture the narrative of both mother and health workers/midwives were taped recorded, transcribed and translated into English. Interviews were conducted at home with mothers and at the field office of VITAL in case of health workers/ midwives. The complete duration for VASA interview was 25–30 minutes and narrative took an additional 20–30 minutes. The quality check of all the VASA forms was performed by senior research coordinator. The dual data entry was done in Redcap. Copies of the medical record (if available) were also attached with the form at the end.

## Cause of death assignment

Physician-coded verbal Autopsy (PCVA) method was used to determine the CoD according to International Statistical Classification of Diseases and Related Health Problems, tenth revision (ICD-10) [32]. A team of four junior physicians and one senior physician were trained by a neonatologist from the Aga Khan University for cause-of-death (CoD) assessment and assignment. The master trainer had received training from the World Health Organization. Each physician received the complete file of VASA paper-based form, translated narrative report and medical record for review; blinded from each other. In case of disagreement in the CoD, the third physician was assigned the same file blinded from the previous assessment. The senior physician was the final decision maker if the agreement is not achieved.

## Data analysis

The analysis of social autopsy variables for neonatal and stillborn deaths was conducted using the STATA 17 software. A descriptive analysis was conducted to present data on stillbirth and neonatal deaths separately according to high- and low-mortality clusters. The chi-square test of independence was conducted to examine the association between categories. Moreover, we assessed pathways to neonatal deaths among the clusters to present patterns of illness recognition and care-seeking. Finally, an in-depth qualitative analysis was conducted. A detailed narrative about the event was used for thematic analysis, which was derived from interviews with mothers and caregivers during the VASA interviews. These narratives were tape-recorded, transcribed into Urdu, and translated into English for analysis. An inductive approach was applied to generate important themes pertaining to stillbirth, neonatal death, and underlying CoDs.

## Ethical considerations

The Ethical Review Committee (ERC) of VITAL Pakistan Trust granted approval for the study (Reference: 001-VPT-IRB-17). Participants gave their written consent in the local language, crafted through the VASA approach, applicable to both retrospective and prospective cases. Written informed consent was taken from the mother/caregiver in a local language. In case

when respondent was not able to read, the research team read the consent form. Participants were encouraged to ask questions regarding the consent form and the VASA interview to ensure their complete understanding. The consent also involved permission to access medical records if available. After consent and permission, medical records were only accessed from the mother or family or the from the partner hospital, Koohi Goth Women Hospital (KGWH). Where required our senior staff discussed the case with hospital KGWH midwife (a partner hospital) and ask for the records or specific variables where family were not able to provide by provided unique ID to the KGWH team.

## Results

### Descriptives

The study captured a total of 421 events. Complete VASA interviews and forms were filled out for 317 (75.3%) of these events. The incomplete VASA interviews accounted for 104 events, divided as follows: 57 interviews (13.5%) were refused, 18 (4.3%) had absent caregivers, and 29 (6.9%) were due to families relocating. Of the completed VASA forms, there were 148 still-births and 169 neonatal deaths. These were further categorized into 192 cases in high mortality clusters and 125 cases in low mortality clusters. (Fig 1).

In the '**high-mortality clusters'**, the VASA was completed for 87 stillbirths and 105 neonatal deaths. Among the stillbirths and neonatal deaths, the mean maternal age was 28.1 ±9.5 years. Regarding ethnicity, 81.8% (157/192) of the participants were Sindhi. In these clusters, 90.6% (174/192) of women had no formal education. Involvement of other family members in the primary decision-makers, accounting for 41.1% (79/192). Regarding toilet facilities, proper flush toilets were available in 62.0% (119/192) households. For household income, poor accounts for 32.3% (62/192) and poorest were 34.9% (67/192). In these clusters, 78.6% (151/192) women reported that they are not using modern contraceptives. Further, 80.7% (155/192) women reported that they have received antenatal care during pregnancy. Although 43.8% of those who sought antenatal care completed an optimal 8 visits and 40.0% mentioned that they were aware of danger signs during pregnancy (Table 1).

In the '**low-mortality clusters'**, the VASA was completed for 61 stillbirths and 64 neonatal deaths. Among the stillbirths and neonatal deaths, the mean maternal age was 29.3 ±6.6 years.

**Figure 1 | Distribution of stillbirth and neonatal death from completed VA**

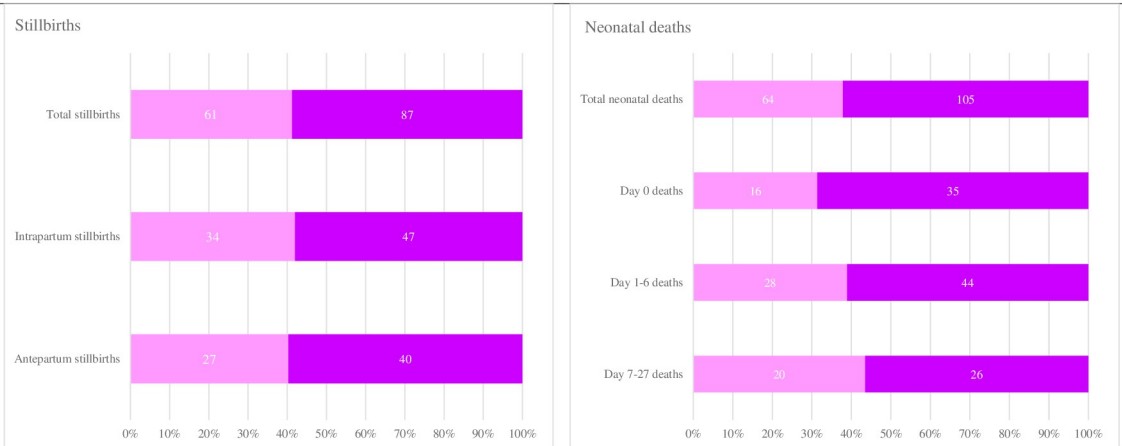

**Fig 1. Distribution of stillbirths and neonatal deaths.** VASA interview successful.

**Table 1. Maternal characteristic for stillbirths and neonatal death cases in Rehri Goth.**

| | High mortality clusters | | | Low mortality clusters | | | Cumulative events of stillbirths and neonatal deaths in Rehri Goth |
|---|---|---|---|---|---|---|---|
| | Stillbirths | Neonatal deaths | Total of stillbirths and neonatal deaths | Stillbirths | Neonatal deaths | Total of stillbirths and neonatal deaths | |
| | N = 87 | N = 105 | N = 192 | N = 61 | N = 64 | N = 125 | 317 |
| **Socio-demographic factors** | | | | | | | |
| Mean maternal age—Mean ±SD | 28.4 ± 5.9 | 29.1 ± 11.7 | 28.8 ± 9.5 | 29.2 ± 6.9 | 29.5 ± 6.4 | 29.3 ± 6.6 | 29.0 ± 8.5 |
| Ethnicity[§] - N (%) | | | | | | | |
| Sindhi | 69 (79.3) | 88 (83.8) | 157 (81.8) | 14 (23.0) | 11 (17.2) | 25 (20.0) | 182 (57.4) |
| Pashto | 13 (14.9) | 7 (6.7) | 20 (10.4) | 16 (26.2) | 16 (25.0) | 32 (25.6) | 52 (16.4) |
| Others | 5 (5.7) | 10 (9.5) | 15 (7.4) | 31 (50.8) | 37 (57.8) | 68 (54.4) | 83 (26.2) |
| Maternal education[§] - N (%) | | | | | | | |
| Primary | 5 (5.7) | 2 (1.9) | 7 (3.6) | 12 (19.7) | 8 (12.5) | 20 (16.0) | 27 (8.5) |
| Secondary | 6 (6.9) | 5 (4.8) | 11 (5.7) | 10 (16.4) | 12 (18.75) | 22 (17.6) | 33 (10.4) |
| No formal education | 76 (87.4) | 98 (93.3) | 174 (90.6) | 39 (63.9) | 44 (68.75) | 83 (66.4) | 257 (81.1) |
| Maternal involvement in decision making at household level—N (%) | | | | | | | |
| Woman herself is the decision maker | 5 (5.7) | 5 (4.8) | 10 (5.2) | 5 (8.2) | 5 (7.8) | 7 (5.6) | 17 (5.4) |
| Husband is the decision maker | 25 (28.7) | 34 (32.4) | 59 (30.7) | 22 (36.1) | 25 (39.1) | 47 (37.6) | 106 (33.4) |
| Mutual decision making with husband | 17 (19.5) | 27 (25.7) | 44 (22.9) | 9 (14.8) | 16 (25.0) | 25 (20.0) | 69 (21.8) |
| Other family members are the decision makers | 40 (46.0) | 39 (37.1) | 79 (41.1) | 25 (40.0) | 18 (28.1) | 46 (36.8) | 125 (39.4) |
| Household density—Mean ±SD | 0.3 ± 0.2 | 0.3 ± 0.4 | 0.3 ± 0.3 | 0.3 ± 0.1 | 0.3 ± 0.1 | 0.3 ± 0.1 | |
| Type of toilet at household[§] - N (%) | | | | | | | |
| Proper flush toilet | 56 (64.4) | 63 (60.0) | 119 (62.0) | 50 (82.0) | 51 (79.7) | 100 (80.0) | 219 (69.1) |
| Improved pit toilet | 11 (12.6) | 22 (21.0) | 33 (17.2) | 5 (8.2) | 2 (3.1) | 6 (4.8) | 39 (12.3) |
| Traditional pit | 16 (18.3) | 13 (12.3) | 29 (15.1) | 5 (8.2) | 9 (14.1) | 14 (11.2) | 43 (13.7) |
| No toilet | 4 (4.5) | 7 (6.7) | 11 (5.7) | 3 (4.9) | 2 (3.1) | 5 (4.0) | 16 (5.0) |
| Income level [φ] - N (%) | | | | | | | |
| Poorest | 28 (32.2) | 39 (37.1) | 67 (34.9) | 17 (27.9) | 12 (18.6) | 29 (23.2) | 96 (30.3) |
| Poor | 35 (40.2) | 27 (25.7) | 62 (32.3) | 25 (41.0) | 27 (42.2) | 52 (41.6) | 114 (36.0) |
| Middle | 13 (14.9) | 11 (10.5) | 24 (12.5) | 10 (16.4) | 12 (18.6) | 22 (17.6) | 46 (14.5) |
| Rich | 1 (1.1) | 14 (13.3) | 15 (7.8) | 5 (8.2) | 5 (7.8) | 10 (8.0) | 25 (7.9) |
| Richest | 6 (6.9) | 3 (2.9) | 9 (4.7) | 2 (3.3) | 4 (6.6) | 6 (4.8) | 15 (4.7) |
| Wealth index cannot be determined | 4 (4.6) | 11 (10.5) | 15 (7.8) | 2 (3.3) | 4 (6.6) | 6 (4.8) | 21 (6.60 |
| Ever used modern contraceptive method—N (%) | | | | | | | |
| Yes | 17 (19.5) | 24 (22.9) | 41 (21.4) | 18 (29.5) | 13 (20.3) | 30 (24.0) | 71 (22.4) |
| No | 70 (80.5) | 81 (71.1) | 151 (78.6) | 43 (70.5) | 51 (79.7) | 95 (76.0) | 246 (77.6) |
| **Care seeking during pregnancy and childbirth** | | | | | | | |
| Sought antenatal care at formal health care provider [§] | | | | | | | |
| Yes | 71 (81.6) | 84 (80.0) | 155 (80.7) | 53 (86.9) | 59 (92.2) | 112 (89.6) | 267 (84.2) |
| No | 16 (18.4) | 21 (20.0) | 37 (19.3) | 8 (13.1) | 5 (7.8) | 13 (10.4) | 50 (15.8) |
| **Number of antenatal visits received** | | | | | | | |

*(Continued)*

**Table 1.** (Continued)

| | High mortality clusters | | | Low mortality clusters | | | Cumulative events of stillbirths and neonatal deaths in Rehri Goth |
|---|---|---|---|---|---|---|---|
| | Stillbirths | Neonatal deaths | Total of stillbirths and neonatal deaths | Stillbirths | Neonatal deaths | Total of stillbirths and neonatal deaths | |
| | N = 87 | N = 105 | N = 192 | N = 61 | N = 64 | N = 125 | 317 |
| *Less than 8 visits* | 45 (51.7) | 63 (60.0) | 108 (56.2) | 30 (49.2) | 36 (56.2) | 66 (52.8) | 174 (54.9) |
| *At least 8 visits* | 42 (48.3) | 42 (40.0) | 84 (43.8) | 31 (50.8) | 28 (43.8) | 59 (47.2) | 143 (45.1) |
| **Awareness received during antenatal visit** | | | | | | | |
| Brief on danger sign during pregnancy [§φ] | | | | | | | |
| *Yes* | 38 (53.5) | 24 (28.6) | 62 (40.0) | 32 (62.7) | 80.7 (54.2) | 64 (58.2) | 126 (47.5) |
| *No* | 33 (46.5) | 60 (71.4) | 93 (60.0) | 19 (37.3) | 27 (45.8) | 46 (41.8) | 139 (52.5) |
| Provider aware woman about, where to go in case of any danger sign [φ] | | | | | | | |
| *Yes* | 47 (66.2) | 29 (34.5) | 76 (49.0) | 33 (64.7) | 31 (52.5) | 64 (58.2) | 140 (52.8) |
| *No* | 24 (33.8) | 55 (65.5) | 79 (51.0) | 18 (35.3) | 28 (47.5) | 46 (41.8) | 125 (47.2) |
| **Place of birth** | | | | | | | |
| *Skilled facility (hospitals or clinics)* | 57 (65.5) | 62 (59.0) | 119 (62.0) | 48 (78.7) | 44 (68.8) | 92 (73.6) | 211 (66.6) |
| *Unskilled (home or informal health facilities)* | 27 (31.0) | 42 (40.0) | 69 (35.9) | 12 (19.7) | 18 (28.1) | 30 (24.0) | 99 (31.2) |
| *On the way to health facility* | 3 (3.4) | 1 (1.0) | 4 (2.1) | 1 (1.6) | 2 (3.1) | 3 (2.4) | 7 (2.2) |
| **Ever breastfed the newborn** | | | | | | | |
| *Yes* | ▓ | 39 (37.1) | 39 (37.1) | 28 (43.8) | 28 (43.8) | 28 (43.8) | 67 (39.6) |
| *No* | ▓ | 66 (62.9) | 66 (62.9) | 36 (56.2) | 36 (56.2) | 36 (56.2) | 102 (60.4) |

§ P-value was <0.05 between "Total of stillbirths and neonatal deaths" among high mortality and low mortality clusters

φ P-value was <0.05 between stillbirths and neonatal deaths within high mortality clusters

Among respondents, Sindhi ethnicity was 57.4% (68/125) and other predominant ethnicities were Pashtun (25.6%), Urdu speaking (16.4%) and Bengalis (11.5%). In these clusters, 64.4% (83/125) of the respondents had no formal education. Other family members were the primary decision-makers, accounting for 36.8% (46/125) of the cases. Access to proper flush toilets was reported by 80.0% (100/125). Income levels showed in these clusters, 41.6% (52/125) and 23.2% (29/125) poor and the poorest, respectively. Women received antenatal care in these clusters were 89.6% (112/125); 47.2% of them attended optimal 8 visits and 58.2% are aware of danger sign during pregnancy (Table 1).

## Cause-of-death

**Antepartum stillbirths.** Overall, pregnancy-induced hypertension contributed to 22.4% of the antepartum stillbirths followed by maternal infections (13.4%), both maternal accidents or injuries resulting in antepartum stillbirth as well as other specific perinatal causes (10.4%) and antepartum hemorrhage (7.5%).

In **'high-mortality clusters'**, pregnancy-induced hypertension (32.5%), maternal infections (22.2%), maternal accidents or injuries resulting in antepartum stillbirth (15.0%), other specific perinatal causes (7.5%) and antepartum hemorrhage (10.0%) were the leading causes. Congenital malformation (5.0%) was determined only in high mortality clusters.

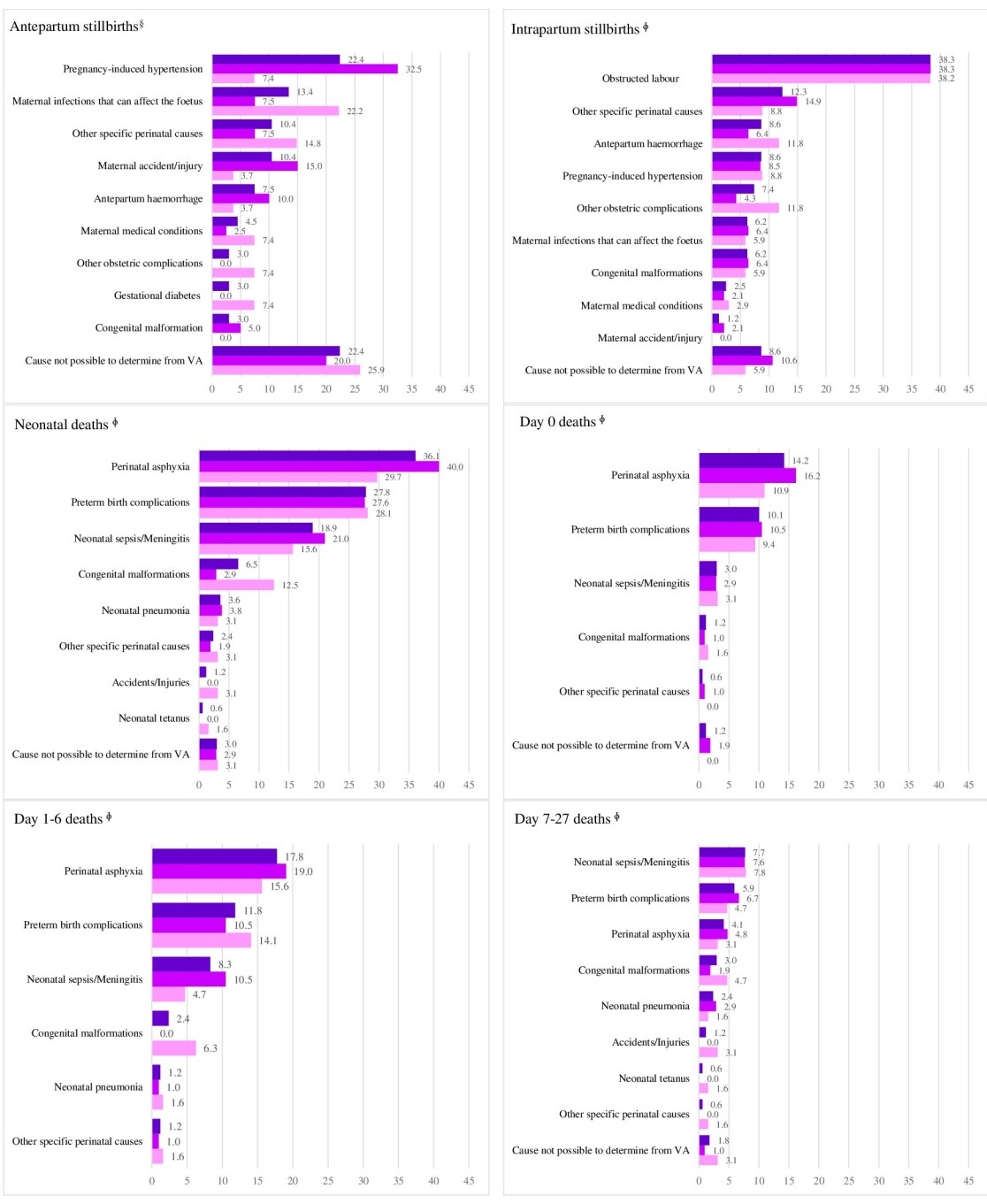

**Fig 2. Cause of stillbirths and neonatal deaths (%).** Segregated by antepartum and intrapartum stillbirths and presented overall as well as age-specific neonatal deaths.

In **'low-mortality clusters'**, maternal infections (22.2%) and other specific perinatal causes (14.8%) were the leading causes. Pregnancy-induced hypertension, maternal medical conditions, other obstetric complications and gestational diabetes were 7.4% each (Fig 2).

**Intrapartum stillbirths.**   Overall, obstructed labor (38.3%), other specific perinatal causes (12.3%), antepartum hemorrhage (8.6%) and pregnancy-induced hypertension (8.6%) were the leading causes for intrapartum stillbirths.

In **'high-mortality clusters'** as well obstructed labor (38.3%) was the leading cause. This is followed by other specific perinatal causes (14.9%), pregnancy-induced hypertension (8.8%) and antepartum hemorrhage (6.4%).

In **'low-mortality clusters'**, after obstructed labor (38.2%) being the leading cause too, antepartum hemorrhage (11.8%) and other obstetric complications (11.8%) were the main causes of intrapartum stillbirths (Fig 2).

**Neonatal deaths.**   Perinatal asphyxia (36.1%), preterm birth complications (27.8%) and neonatal sepsis/meningitis (18.9%) are the leading overall neonatal causes. Further subgroup analysis revealed that among day 0 as well as day 1–6 deaths, perinatal asphyxia and preterm birth complications remained the top two leading causes in those age groups. For age 7–27 days, neonatal sepsis/meningitis and preterm birth complications were leading causes.

In **'high-mortality clusters'**, perinatal asphyxia (40.0%), preterm birth complications (27.6%) and neonatal sepsis/meningitis (21.0%) are the leading overall neonatal causes.

In **'low-mortality clusters'**, the leading causes were perinatal asphyxia (29.7%), preterm birth complications (28.1%) and neonatal sepsis/meningitis (15.6%). Congenital anomalies (12.5%) also among leading causes (Fig 2).

## Pathways to neonatal deaths

Overall, illness recognition and the consequent care-seeking pathways did not exhibit significant differences among the identified clusters. Initial recognition of fatal illnesses was primarily conducted by medical professionals, including doctors or nurses (46.2%, 78/169) and traditional birth attendants (11.8%, 20/169). Of the total 144 cases where care was sought, 52.1% (N = 75) were referred to another facility. The remaining 47.9% (N = 69) were managed at the initial facility, and among them, 17.4% (N = 25) succumbed to the illness. Additionally, 15.2% (N = 22) left the first facility and passed away at home, while N = 17 left the first facility against medical advice and sought care at a second facility. In total, 97 newborns sought care at the second health facility, where 9.3% (N = 9) were subsequently referred to a third health facility. Moreover, 72.2% (N = 70) of these newborns died at the second health facility, while 27.8% (N = 27) left the facility alive but later died either at home or another facility (Table 2).

**In 'high-mortality clusters"**, the perception that care is not needed was reported by 11.4% and in 86.7% of cases care was sought or received.

In **'low-mortality cluster'**, there was no case where care was considered unnecessary.

## Thematic analysis

The purposive sample of 40 detailed narratives and qualitative data showed that health system (62.5%) and community factors (37.5%) were contributing root causes of stillbirths and neonatal deaths. In-depth analysis showed that within health system related factors (65%, 13/20) in **'high-morality clusters'**, the delayed referral (46.2%), suboptimal quality of care (30.8%), and poor triage system (23.1%) were the main factors which lead to the specific cause of fatal events (Fig 3).

Delayed referrals were noted in cases of antepartum stillbirths, particularly due to antepartum hemorrhage and pregnancy-induced hypertension

*"I waited in clinic for almost 2.5 hours. I was bleeding and my mother was requesting them again and again, they said the vehicle is coming, but it came very late." (Antepartum stillbirth, high mortality clusters, mother).*

**Table 2. Pathways of care seeking during fatal illness among newborns.**

| Pathways | High mortality clusters | Low mortality clusters | Total neonatal deaths in Rehri Goth | p-value |
|---|---|---|---|---|
| | N = 105 | N = 64 | N = 169 | |
| Initial recognition of fatal illness–N (%) | | | | 0.92 |
| Mother | 27/105 (25.7) | 17/64 (26.6) | 44/169 (26.0) | |
| Other relative, neighbor, friend | 17/105 (16.2) | 8/64 (12.5) | 25/169 (14.8) | |
| Health worker visiting home | 1/105 (1.0) | 1/64 (1.6) | 2/169 (1.2) | |
| Doctor or nurse at a health facility | 49/105 (46.7) | 29/64 (45.3) | 78/169 (46.2) | |
| Other provider, like traditional birth attendant | 11/105 (10.5) | 9/64 (14.1) | 20/169 (11.8) | |
| Any care sought or received, or treatment given or received by newborn–N (%) | | | | 0.005 |
| Yes | 91/105 (86.7) | 59/64 (92.2) | 150/169 (88.8) | |
| No (perceived that care not needed, given, or sought) | 12/105 (11.4) | 0/64 (0.0) | 12/169 (7.1) | |
| No (died immediately) | 2/105 (1.9) | 5/64 (7.8) | 7/169 (4.1) | |
| First action which was taken immediately after recognition of fatal illness–N (%) | | | | 0.52 |
| Home care provided | 11/91 (12.1) | 4/59 (6.8) | 15/150 (10.0) | |
| Traditional care provided | 0/91 (0.0) | 2/59 (3.4) | 2/150 (1.3) | |
| Self-medication given, purchased from pharmacy or drug seller | 1/91 (1.1) | 0/59 (0.0) | 1/150 (0.7) | |
| Care sought at private clinic | 11/91 (12.1) | 7/59 (11.9) | 18/150 (12.0) | |
| Care sought at some governmental or non-governmental clinic | 9/91 (9.9) | 5/59 (8.5) | 14/150 (9.3) | |
| Care sought at hospital | 6/91 (6.6) | 3/59 (5.1) | 9/150 (6.0) | |
| Illness began at provider where newborn was delivered | 53/91 (58.2) | 38/59 (64.4) | 91/150 (60.7) | |
| Referral to another health provider or facility from first health provider*–N (%) | | | | 0.54 |
| Yes | 43/86 (50.0) | 32/58 (55.2) | 75/144 (52.1) | |
| No | 43/86 (50.0) | 26/58 (44.8) | 69/144 (47.9) | |
| Vital status of newborn before leaving first health provider–N (%) | | | | 0.98 |
| Yes, left alive and died later at another facility or home | 71/86 (82.6) | 48/58 (82.8) | 119/144 (82.6) | |
| No, died at this provider | 15/86 (17.4) | 10/58 (17.2) | 25/144 (17.4) | |
| Any advice given at first health provider before leaving–N (%) | | | | 0.79 |
| Yes | 66/71 (93.0) | 44/48 (91.7) | 110/119 (92.4) | |
| No | 5/71 (7.0) | 4/48 (8.3) | 9/119 (7.6) | |
| Able to follow advice–N (%) | | | | 0.72 |
| Yes | 50/66 (75.8) | 32/44 (72.7) | 82/110 (74.5) | |
| No | 16/66 (24.2) | 12/44 (27.3) | 28/110 (25.5) | |
| Referral to another health provider or facility from second health provider–N (%) | | | | 0.18 |
| Yes | 7/55 (12.7) | 2/42 (4.8) | 9/97 (9.3) | |
| No | 48/55 (87.3) | 40/42 (95.2) | 88/97 (90.7) | |
| Vital status of newborn before leaving second health provider–N (%) | | | | 0.092 |
| Yes, left alive | 19/55 (34.5) | 8/42 (19.0) | 27/97 (27.8) | |
| No, died at this provider | 36/55 (65.5) | 34/42 (81.0) | 70/97 (72.2) | |
| Any advice given at second health provider before leaving $ –N (%) | | | | 0.77 |
| Yes | 13/19 (68.4) | 5/8 (62.5) | 18/27 (66.7) | |
| No | 6/19 (31.6) | 3/8 (37.5) | 9/27 (33.3) | |

ᶲ Denominator in different categories changes, we are reporting respective denominator

ᵠ Denominator is those who accepted the referral

$ Out of total 27 who were referred to third facility, 6 were referred to third facility and died there or at home

Similarly, in intrapartum cases, poor triage systems and delayed referrals led to stillbirths, primarily caused by obstructed labor. This was even more common for primigravida women.

"*More than 24 hours had passed (at facility), and there was nothing. It was a Sunday, and no seniors were there. I was in pain and shouting, but they kept saying, wait. Then, suddenly, we were referred to another hospital. They operated on me and said the baby is dead already.*" *(Intrapartum stillbirth, high mortality clusters, mother)*

A women shared the experience with neonatal death as:

"*I was perfectly fine throughout the pregnancy; this was my first child. They performed C-section the next day. When my baby was born, she did not cry, they gave her treatment and asked my family to take her to another hospital. She died as soon as she got there.*" *(Day 0 death, high mortality clusters, mother)*

With regards to community factors (35.0%, 7/20) in **'high-mortality clusters'**, delay in care seeking was the leading aspect into specific causes (Fig 3).

"*We thought that our Dai (traditional birth attendant) would solve the problem. I delivered the child with the same Dai before she became experienced. She tried hard but decided to call*

**Fig 3: Thematic analysis of root causes §**

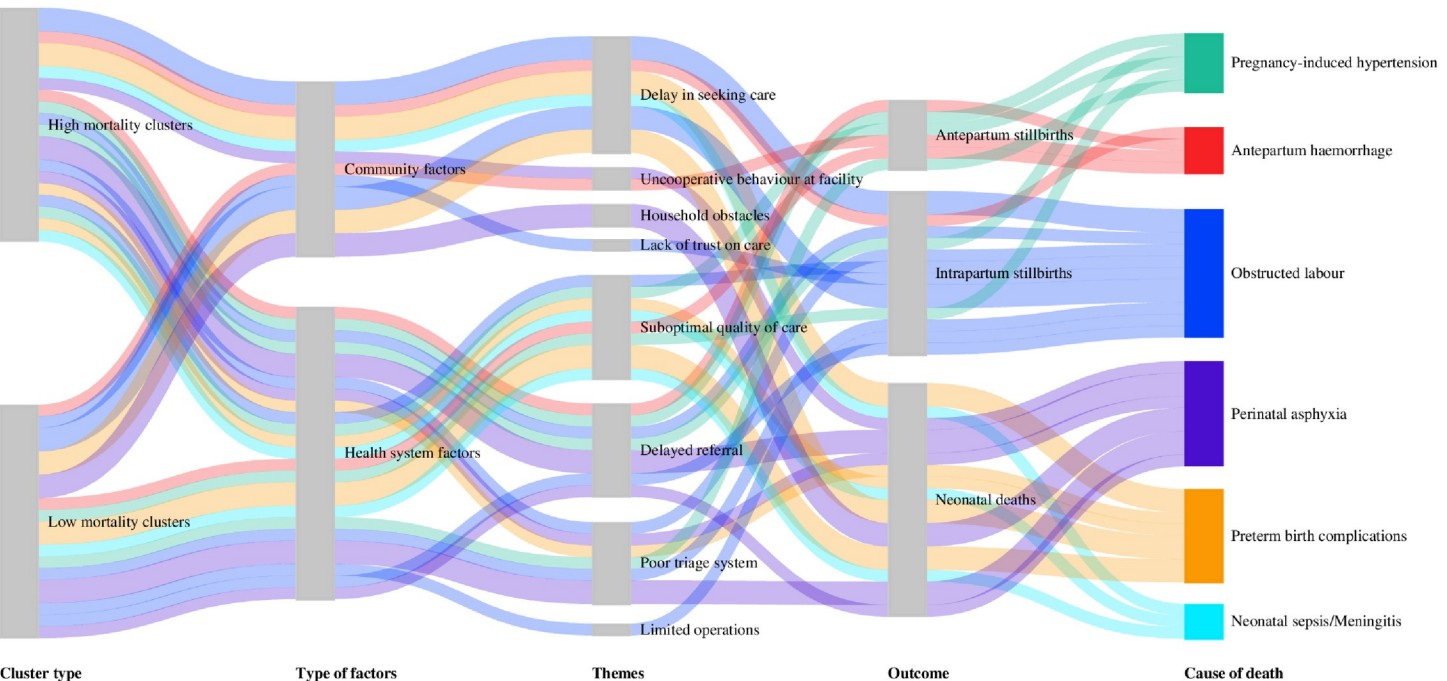

§ *Unique color code is assigned to specific cause of antepartum and intrapartum stillbirths and neonatal death among selected cases which traced back to the themes of root causes, type of factors and cluster type.*

**Fig 3. Thematic analysis of root causes.** Health systems and community factors.

*the center for a checkup. They said I am in labor, but the baby seemed already dead."* (Intra-partum stillbirth, high mortality clusters, mother)

In **'low-mortality clusters'** also, the contributing factors toward the fata event were health system related (65.0%, 12/20); under which suboptimal quality of care (41.7%) and poor triage (33.3%) were the main identifies reasons (Fig 3).

One of the mothers-in-law shared her experience with the maternal death and intrapartum stillbirths as:

*"In the last days, she had difficulty breathing and the [government hospital] staff didn't pay attention. Otherwise, she would have survived. They wasted too much time. If we had known, we would have taken her to a private facility. . .she never had any problem other than diffi-culty breathing. . . I would not recommend anyone to go [to the government hospital]. (Intra-partum stillbirth and maternal death, low mortality clusters, care coordination midwife)*

A woman shared her experience of antepartum stillbirth:

*"My blood pressure was high, and I was regularly visiting for a checkup. During my last two visits, I complained of feeling less movement from the baby, but the nurse did not listen to me. They did not inform me about what is happening to me, and were saying that you are fine"* (Antepartum stillbirth, low mortality clusters, mother)

Among community factors (40.0%,8/20) in low-mortality clusters, delay in care seeking (33.3%) was the leading obstacle to CoD.

A woman shares the reason of delay as:

*"When the baby was born (at home), at that time, I had severe bleeding. At birth, the baby was very weak. He did cry but had difficulty breathing. . . . . . TBA told us to immediately take him (to a private clinic). However, just arranging the car and everything, it took two hours. The doctor said that he had already died on the way."* (Neonatal death, low mortality clusters).

## Discussion

The finding revealed that 45.3% of the stillbirths were during the antepartum period compared to 54.7% intrapartum. A report from the Aga Khan University from Rehri Goth and other neighboring communities showed 83% of antepartum and 17% of intrapartum stillbirths in 2012 [33]. The same data suggested that pregnancy-induced hypertension (37%), antepartum hemorrhage (10%) and obstructed labor (6%) were the top three causes of stillbirths in these areas [33]. However, our study showed that obstructed labor is the top leading cause (20.9%) followed by pregnancy-induced hypertension (14.9%), other specific perinatal causes (11.5%) and maternal infections (9.5%). The difference in distribution and burden of causes can be explained by the availability of integrated MNCH services at Rehri Goth since 2014, and com-prehensive data collection of these events at partner facility to ascertain the causes.

Specific to the causes of stillbirths, pregnancy-induced hypertension (22.4%), which was higher in high-mortality clusters, maternal infections (13.4%), other perinatal causes (10.4%), and maternal accidents (10.4%) shared more than half of the burden of antepartum stillbirths in Rehri Goth. CoD for antepartum stillbirth were significantly different between the low-and high-mortality clusters. Moreover, obstructed labour (38.3%), other perinatal causes (12.3%),

antepartum hemorrhage (8.6%), and pregnancy-induced hypertension (8.6%) contributed to two-thirds of intrapartum stillbirths. The CoD analysis clearly suggests that the quality of obstetric care and quality healthcare is crucial during both antenatal and intrapartum care. Such provision of care in a resource poor setting is difficult to access. Consequently, this resulted in fatal events and left a huge burden on both the health system and maternal health. The burden of these preventable causes from the South Asian region showed that hypertensive disorder during pregnancy (47%), antepartum hemorrhage (21%) and maternal infections are the leading causes [15]. The data in this study included peri-urban slums of Karachi, including Rehri Goth, but the CoDs for stillbirths are reported by region. The same study reported 57% of intrapartum stillbirths due to complications of labor and delivery and 20% due to hypertensive disorders during pregnancy [15].

It is crucial to interpret these findings within the context of the study's limitations, which include the reliance on retrospective data and potential recall bias. However, it is important to highlight that the VASA data also encompasses prospective deaths, thus mitigating some of the recall bias concerns. Another limitation could be the inclusion of only 40 narratives and qualitative data in the analysis based on convenient sampling of the already collected data. However, this helped us in performing robust data triangulation using information from multiple sources, including medical records, narratives of health workers, and care coordination midwives. This approach enabled a comprehensive understanding of various factors and causes of death, a level of detail not explicitly reported in previous VASA studies. Despite these limitations, the study's results offer valuable insights into the underlying causes, contributing factors, and pathways leading to stillbirths and neonatal deaths in the community of Rehri Goth

Furthermore, with regards to neonatal deaths, more than three fourths of them were linked to more preventable causes like perinatal asphyxia, preterm birth complications and neonatal sepsis or meningitis. Neonatal deaths were primarily attributed to perinatal asphyxia (36.1%), with slightly higher rates observed in high-mortality clusters (40.0%). Preterm birth complications (27.8%) were also identified as a significant cause of neonatal deaths, with some variations among clusters. A previous study from the Rehri Goth and adjacent peri-urban slums showed that 40% of the neonatal deaths were due to perinatal asphyxia followed by preterm birth complications (23.0%) [34]. Despite similar results, our study demonstrated neonatal sepsis or meningitis as the CoD in 18.9% cases compared to 29.0% in the previous study [34].

Examining the sociodemographic factors associated with stillbirths and neonatal deaths provides valuable insights into the context of these adverse outcomes. The study identified variations in ethnicity, maternal education, and income levels between high- and low-mortality clusters. The ethnic composition of the clusters also influenced healthcare outcomes, with Sindhi ethnicity dominating both clusters, but greater ethnic diversity in the low mortality clusters. These findings highlight the importance of contextually tailored interventions to address the specific needs and challenges faced by different ethnic groups [35]. Our study revealed a higher proportion of mothers without formal education in high-mortality clusters, underscoring the need for targeted interventions to improve education and awareness regarding maternal and child health. Additionally, income levels showed no significant differences between high- and low-mortality clusters, but most events occurred in the poorest and poor income categories [36]. Within the high mortality clusters, stillbirths were predominantly observed in the poorest quintiles, indicating a socioeconomic gradient in adverse outcomes [37]. These findings emphasize the importance of addressing socioeconomic disparities and implementing interventions that are accessible and affordable for the most vulnerable populations [38].

The pathways to neonatal deaths identified in this study provide valuable insight into the healthcare-seeking behaviors and care delivery processes. The study found that illness

recognition pathways were not significantly different among clusters, with healthcare professionals and traditional birth attendants being the primary sources of initial recognition [39]. These findings are very different from previously reported pathways in other LMICs using the same VASA tool [6, 28, 40]. However, care-seeking patterns differed between high- and low-mortality clusters, with a higher proportion of cases perceiving care was not needed in high mortality clusters [41]. This highlights the importance of addressing misconceptions and barriers to seeking care in high-mortality clusters. Furthermore, our study revealed challenges in the referral system, with delayed referrals and poor triage systems leading to adverse outcomes. Improving the quality and efficiency of referral processes, as well as strengthening healthcare facilities' capacity to provide appropriate care, are critical interventions to prevent neonatal deaths [42]. In many cases of stillbirths as well as neonatal deaths, a well-equipped primary health care facility can avert the risk of poor fatal outcomes [43].

In many resource-constraint settings the community and health system factors contribute to the suspected root causes of stillbirths and neonatal deaths [44]. Delayed referrals and poor triage systems were identified as health system factors influencing adverse outcomes, particularly in cases of antepartum stillbirths and intrapartum stillbirths caused by obstructed labor [45]. The suboptimal quality of care within the health system also contributed to adverse outcomes in intrapartum cases [46] [47, 48]. This is even more crucial for health facilities where midwives are involved as frontline staff in providing antenatal and intrapartum care [49]. Community factors, such as delay in seeking care and lack of trust in healthcare facilities, were also significant contributors to adverse outcomes, particularly in cases of perinatal asphyxia and preterm birth complications [50]. Addressing both health systems and community factors through comprehensive interventions and community engagement to improve maternal and neonatal health outcomes is pivotal to address the underline causes [51].

The findings underscore the urgent need for targeted interventions that address both individual and systemic factors to effectively reduce mortality rates in this population. To further advance our knowledge in this field, future research should focus on evaluating the effectiveness of specific interventions and strategies aimed at tackling the identified factors. By doing so, we can foster improvements in maternal and child health outcomes, leading to better health for the community as a whole. It is imperative to use this evidence to inform evidence-based policies and programs that will ultimately benefit the most vulnerable members of society.

## Conclusion

The findings generated in-depth understanding with regards to the causes of stillbirths and neonatal deaths in Rehri Goth, a context where comprehensive MNCH program was implemented. Further, deep insights by segregating by clusters as well as triangulation with qualitative analysis underscored the needs of evidence-based strategies for MNCH interventions in impoverished community. Specific to the causes such as pregnancy-induced hypertension, maternal infections, obstructed labor, perinatal asphyxia, and preterm birth complications, these findings necessitate targeted interventions, including improved antenatal care, skilled birth attendance, emergency obstetric services, and neonatal resuscitation and intensive care. Continued research and evaluation of these interventions is essential to monitor their effectiveness and make informed changes as needed which is crucial to make progress towards reducing stillbirths and neonatal deaths and improving the overall well-being of mothers and newborns in Rehri Goth.

## Supporting information

**S1 Data.**
(DTA)

**S2 Data.**
(DTA)

**S1 File.**
(PDF)

## Author Contributions

**Conceptualization:** Ameer Muhammad, Yasir Shafiq.

**Data curation:** Ameer Muhammad.

**Formal analysis:** Ameer Muhammad, Yasir Shafiq.

**Funding acquisition:** Yasir Shafiq.

**Investigation:** Yasir Shafiq.

**Methodology:** Ameer Muhammad.

**Project administration:** Ameer Muhammad, Yasir Shafiq.

**Resources:** Yasir Shafiq.

**Software:** Ameer Muhammad, Alishan Bachlany.

**Supervision:** Ameer Muhammad, Benazir Baloch, Yasir Shafiq.

**Validation:** Ameer Muhammad.

**Visualization:** Alishan Bachlany.

**Writing – original draft:** Ameer Muhammad, Muhammad Salman Haider Rizvee, Uzma Khan, Hina Khan, Alishan Bachlany, Benazir Baloch, Yasir Shafiq.

**Writing – review & editing:** Ameer Muhammad, Muhammad Salman Haider Rizvee, Benazir Baloch, Yasir Shafiq.

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
