## [Decision Letter · Decision Letter 0]

16 Oct 2023

PONE-D-23-24729Uncovering the Causes and Socio-demographic Constructs of Stillbirths and Neonatal Deaths in an Urban Slum of KarachiPLOS ONE

Dear Dr. Muhammad,

Thank you for submitting your manuscript to PLOS ONE. After careful consideration, we feel that it has merit but does not fully meet PLOS ONE’s publication criteria as it currently stands. Therefore, we invite you to submit a revised version of the manuscript that addresses the points raised during the review process.

We look forward to receiving your revised manuscript.

Kind regards,

Sidrah Nausheen, FCPS

Academic Editor

PLOS ONE

Journal Requirements:

2. Thank you for submitting the above manuscript to PLOS ONE. During our internal evaluation of the manuscript, we found significant text overlap between your submission and previous work in the [introduction, conclusion, etc.].

Please revise the manuscript to rephrase the duplicated text, cite your sources, and provide details as to how the current manuscript advances on previous work. Please note that further consideration is dependent on the submission of a manuscript that addresses these concerns about the overlap in text with published work.

[If the overlap is with the authors’ own works: Moreover, upon submission, authors must confirm that the manuscript, or any related manuscript, is not currently under consideration or accepted elsewhere. If related work has been submitted to PLOS ONE or elsewhere, authors must include a copy with the submitted article. Reviewers will be asked to comment on the overlap between related submissions (http://journals.plos.org/plosone/s/submission-guidelines#loc-related-manuscripts).]

We will carefully review your manuscript upon resubmission and further consideration of the manuscript is dependent on the text overlap being addressed in full. Please ensure that your revision is thorough as failure to address the concerns to our satisfaction may result in your submission not being considered further.

5. Please amend your list of authors on the manuscript to ensure that each author is linked to an affiliation. Authors’ affiliations should reflect the institution where the work was done (if authors moved subsequently, you can also list the new affiliation stating “current affiliation:….” as necessary).

Reviewers' comments:

Reviewer's Responses to Questions

**Comments to the Author**

1. Is the manuscript technically sound, and do the data support the conclusions?

Reviewer #1: Partly

2. Has the statistical analysis been performed appropriately and rigorously? 

Reviewer #1: No

3. Have the authors made all data underlying the findings in their manuscript fully available?

Reviewer #1: No

4. Is the manuscript presented in an intelligible fashion and written in standard English?

Reviewer #1: No

5. Review Comments to the Author

Reviewer #1: PONE-D-23-24729

Uncovering the Causes and Socio-demographic Constructs of Stillbirths and Neonatal Deaths in an Urban Slum of Karachi

Summary

This study emphasizes the need to address stillbirths and neonatal deaths in Pakistan, with high rates of 43 per 1,000 births and 46 deaths per 1,000 live births. Efforts should focus on improving access to quality antenatal care, skilled birth attendants, and healthcare infrastructure, particularly in disadvantaged areas and urban slums. Addressing cultural norms, health-seeking behaviors, and resource allocation is essential for reducing stillbirths and neonatal deaths and improving the well-being of mothers and newborns. The study's main goals are to identify the causes of stillbirths and neonatal deaths in an urban slum named “Rehri Goth”, analyze relevant socio-demographic characteristics, and investigate community and health system factors.

Abstract:

- Please add a line on pathways leading to stillbirth and neonatal deaths in the introduction.

- Please mention the study design for clarification.

- Please add a line about data analysis in the methods

- It is unclear whether this was a primary study or a secondary analysis.

- It is unclear who reviewed verbal and social autopsies and how scoring was done.

- It is worth adding percentages for health system and community factors.

- Please write the conclusion of your study.

Introduction:

Overall, the introduction is good, However, the text can benefit from simplifying sentences, reorganizing information for better flow, and reducing redundancies to enhance readability and clarity.

Methods:

- Please mention the study design (Seems like a mixed method/multimethod study)

- Too many dates, I suggest restricting the date to the date of this study.

- Please mention in a paragraph about the MNCH program in the setting.

- Sample size assumptions are missing.

- Please specify why 40 narrative interviews for thematic analysis were chosen.

- Is the VASA tool validated?

- What was the sampling technique?

- The data collection method should be elaborated rather than details of training of staff.

- The first six lines of the “Data analysis” section should be moved to the methodology section.

- Please describe the interview guides and how the participants identified and approached each other.

- It is worth adding details about “Koohi Goth Women's Hospital”, especially in terms of maintaining medical records, and the record retrieval process.

- The section on “Ethical consideration and informed consent procedure” should be simplified in terms of language, and fluency for readability and clarification.

Results:

- The first paragraph “Descriptives” needs to be rephrased and avoid long sentences that are not connected for clarification.

For example: “A total of 421 events were captured; a complete VASA interview was conducted, and forms were filled for 317 (75.3%) events. Out of 104 with incomplete VASA interview status; interviews refused were 57 (13.5%), caregivers were absent were 18 (4.3%), and shifted were 29 (6.9%).”

- I suggest breaking the results into two main sections: 1) high mortality cluster and 2) low mortality cluster for clarification and readability and easy to follow.

- In the results section present the results only not elaborating or interpreting the findings

- For privacy and confidentiality, don’t mention the name of the health facility.

- Quotes used in the results are too long to follow and have little meaning conveyed.

Discussion

- The first 3 lines should be removed because of repitation.

- Write strengths and limitations in the second paragraph.

- The conclusion in the discussion should be in sync with the conclusion in the abstract.

Figures:

- Please add the legend (What color code indicated)

- There are three figures, but the caption of figure is missing.

- What is the purpose of figure three? Should be described in either result od discussed in the discussion section?

6. PLOS authors have the option to publish the peer review history of their article (what does this mean?). If published, this will include your full peer review and any attached files.

Reviewer #1: No

---

## [Author Response · Author response to Decision Letter 0]

20 Dec 2023

Respected editors and reviewers, 

Many thanks for your constructive feedback, it helped us alot in improving the content and reporting of our key findings. We have addressed all the comments shared and uploaded the point-by-point response. 

Please let us know in case of further questions.

Regards

Ameer

---

## [Editor Report · Decision Letter 1]

20 Jan 2024

Uncovering the Causes and Socio-demographic Constructs of Stillbirths and Neonatal Deaths in an Urban Slum of Karachi

PONE-D-23-24729R1

Dear Dr. Ameer Muhammad,

We’re pleased to inform you that your manuscript has been judged scientifically suitable for publication and will be formally accepted for publication once it meets all outstanding technical requirements.

Kind regards,

Sidrah Nausheen, FCPS

Academic Editor

PLOS ONE
---

## [Editor Report · Acceptance letter]

27 Mar 2024

PONE-D-23-24729R1 

PLOS ONE

Dear Dr. Muhammad, 

I'm pleased to inform you that your manuscript has been deemed suitable for publication in PLOS ONE. Congratulations! Your manuscript is now being handed over to our production team.

Kind regards, 

on behalf of

Dr. Sidrah Nausheen 

Academic Editor

PLOS ONE